# Molecular epidemiology of large dengue serotypes outbreak from Peru, 2023

**Orson Mestanza[1]/+, Wendy Lizarraga[1], Princesa Medrano[1], María P García[2], Susy Merino[2], Iris S Molina[1], Nieves Sevilla[1], Veronica Hurtado[1], Luis Barcena-Flores[1], Vanesa Izarra-Rojas[1], Karla Vásquez-Cajachahua[1], Alicia Nuñez-Llanos[1], Carlos Padilla-Rojas[1], Henri Bailon[1], Marco Galarza-Perez[1], Dana Figueroa[2], Adolfo Marcelo[2], Victor Jimenez-Vasquez[1]**

[1]National Centre for Public Health, National Institute of Health, Area of Innovation and Development, Lima, Peru
[2]National Centre for Public Health, National Institute of Health, National Metaxenic and Viral Zoonoses Reference Laboratory, Lima, Peru

**BACKGROUND** Dengue has long been considered a neglected disease since its emergence. However, in recent years, the number of cases has increased significantly, particularly in tropical countries across Latin America.

**OBJECTIVE** This study aims to describe the evolution of dengue during the 2023 outbreak in Peru.

**METHODS** Here, we describe the analysis of 245 positive samples for dengue carried out by the National Institute of Health (INS) of Peru through genomic surveillance protocols, and analyse the evolution of serotypes and genotypes during the outbreak.

**FINDINGS** We analysed 245 dengue-positive samples sequenced by the INS of Peru during the 2023 outbreak to describe the evolutionary dynamics of circulating serotypes. We identified dengue virus (DENV)-1 genotype V (lineages D.1 and D.2), DENV-2 genotype II (Cosmopolitan, lineage F.1.2), and DENV-3 genotype III (lineage B.3). Phylogenetic and population-structure analyses revealed multiple introductions, rapid diversification, and wide dispersal of DENV-1 and DENV-2 across Peru, while DENV-3III showed a localised introduction in Lima with phylogenetic links to the Caribbean. Our findings provide the most comprehensive genomic characterisation to date of dengue circulation in Peru in 2023 year, and highlight the need for sustained genomic surveillance.

**MAIN CONCLUSION** To summarise, the three serotypes of dengue in Peru show a dynamic diversity that becomes necessary to understand, to improve public health decisions by authorities.

Key words: dengue virus - genotypes - whole genome sequencing - 2023 outbreak - Peru

Dengue figever has become a global health concern due to its higher incidence, possibly linked to vector migration, frequent international travel, urban growth, deforestation, and climate change. It is estimated to be prevalent in around 100 countries, impacting 96 million people each year.[1] It is transmitted mainly by the mosquito vector *Aedes aegypti*, however, some studies suggest *Aedes albopictus* as a secondary vector.[2] It is well known that dengue is passed on in nature by two distinct cycles, the first is the urban cycle, in which humans and mosquitoes spread the disease, and the second is the sylvatic cycle, transmitted by non-human primates and mosquitoes.[3] Historically, the dengue virus (DENV) is classified into four serotypes (DENV-1, DENV-2, DENV-3, and DENV-4), which are found in Africa, Asia, and the Latin America continent.[4] However, a new fifth serotype (DENV-5) was found and proposed in Malaysia in 2007.[5] Dengue classification includes several genotypes inside each serotype. Among them, DENV-1 is divided into five genotypes (I-V), DENV-2

into six genotypes (I-VI), DENV-3 into four genotypes (I-IV), and DENV-4 into four genotypes (I-IV). Recently, in September 2024, an international team of researchers proposed a classification system for dengue diversity (https://dengue-lineages.org/lineages.html), defining a nomenclature based on a major and minor lineage. This organisation represents the first improvement in the proposal of new lineages and allows traceability of those that represent a public health concern.[6]

Since the severe acute respiratory syndrome coronavirus 2 (SARS-CoV-2) pandemic, several countries have established or enhanced genomic surveillance, making it possible to quickly understand how viral diversity can enhance immunological escape and utilise the information obtained to develop viral disease diagnostics and prepare treatments for the near future.[7] In a sense, including genomic surveillance of arboviruses like those previously reported from Ecuador, Brazil, and Sri Lanka becomes extremely important, especially for endemic countries due to their higher incidence.

**doi:** 10.1590/0074-02760250204

**Financial support:** This research has used financial resources from the Government of Peru through the budget allocated to the Ministry of Health and the National Institute of Health.
**+ Corresponding author:** omestanza@ins.gob.pe | ⓘ https://orcid.org/0000-0001-7268-0496

**Handling editor:** Alexandre J da Silva | ⓘ https://orcid.org/0000-0002-0865-6421

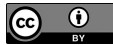

[8] Of the 9.1 million dengue cases documented in all Latin American countries between 2019 and 2022, only 367 DENV genomes (> 70% sequencing coverage) have been published in the GenBank database.[9] In Peru, the first reported cases of dengue were found in the Amazon Peruvian (Loreto, region of Peru) in 1990.[10] Since then, 315 Peruvian outbreaks were detected between 1996 and 2006, highlighting the years 1996 and 2001, since they were epidemics with more than 1,500 positive cases.[11] Wide dengue outbreaks were reported in 2011, 2013, and 2015, with a massive number of positive cases exceeding 20,000 in 2017.[12] The recent large outbreak of dengue in Peru started at the end of 2022 and continued until the middle of 2023. It was caused by the spread of three serotypes of dengue (DENV-1, DENV-2, and DENV-3) with 115,949 reported cases until the 21st epidemiological week.[13]

The principal challenge with the DENV until now is the lack of cross-protection against other serotypes after infection. Researchers suggest that a primary infection could be asymptomatic or cause mild symptoms, while a second infection causes severe disease.[14] Due to the limited or lack of genomic information on different outbreaks of the past, it is difficult to understand the complete virus evolution locally. In addition, during the beginning of 2023, the Peruvian climate was disturbed by the Yaku cyclone and the subsequent El Niño phenomenon, promoting the dispersion and transmission of dengue throughout Peruvian territory and recording an unprecedented increase in cases. Despite recurrent dengue epidemics in Peru, comprehensive genomic datasets characterising the evolutionary history and spread of DENV serotypes have been scarce. This study provides a detailed genomic reconstruction of the 2023 outbreak, focusing on the population dynamics, introduction events, and dispersal patterns of DENV-1, DENV-2, and DENV-3.

## MATERIALS AND METHODS

*RNA extraction and sequencing* - Samples were collected by several Regional Reference Laboratories and sent to the Metaxenic and Viral Zoonoses Laboratory of the National Institute of Health (INS) in Peru to perform diagnoses and sequencing. Here, we evaluated 245 samples with positive diagnoses for dengue serotype collected from January to June 2023. The DENV RNA extraction of human serum samples was performed using the QIAamp Viral RNA Kit (QIAGEN) following the manufacturer's instructions. Serum sample extraction and library preparation were conducted on the same working day because dengue RNA is very unstable and susceptible to fragmentation. The genomic libraries were prepared according to the Nextera XT protocol, which has the following stages: cDNA preparation, amplification, and pooling of libraries. For cDNA amplification two groups of primers by serotype proposed in the "CDC Next Generation Sequencing Protocol for DENV-1-4 using Illumina Miseq" made by the Centre for Disease Control and Prevention (CDC) were used. Finally, whole genome sequencing was conducted on the Miseq sequencer (Illumina, Inc).

*Genome analysis* - The quality of sequencing raw data was tested using the tool FastQC v0.11.9.[15] To obtain genomes we decided to index and map reads against a genome reference through software packages like BWA v0.7.17[16] and Samtools v1.13.[17] The genome consensus for each sample was constructed using the program IVAR v1.3.1 with the parameters -q 25, -t 0.6, and -m 5.[18] It is necessary to detail that the serotype was identified before sequencing by serological test and confirmed by Real-time polymerase chain reaction (qRT-PCR), while genotype and lineage identification were performed after sequencing using the online tool Genome Detective.[19]

For phylogenetics and population analysis performance, a dataset of DENV genomes (> 8000 bp) was downloaded from the GenBank database,[20] with a minimum of complete data for the year and month in the sample isolation section for DENV-1, DENV-2, and DENV-3. Additionally, recent arbovirus databases of Global Initiative on Sharing All Influenza Data (GISAID) were downloaded with a complete isolation date for all serotypes. The genome references used for each serotype (DENV-1, DENV-2, DENV-3) were collected from the GenBank database under the accession numbers NC_001477, NC_001474, and NC_075435, respectively. The phylogenetic trees were constructed by serotype using the Nextstrain program and the IQ-TREE method.[21]

*Inference of genetic structure* - To understand the genetic composition and population structure of Peruvian dengue serotypes a Bayesian analysis was carried out, with Bayesian evolutionary analysis by sampling trees (BEAST) v10.5.0.[22] The clades containing the ancestors of DENV-1V (364 genomes) and DENV-2II (316 genomes) were selected and analysed in the population Structure v.2.3.4.[23] Genomes were aligned with the respective references and single informative nucleotide polymorphisms were detected with MEGA6.[24] Then, we converted multi-FASTA to VCF using PGDSpider.[25] For DENV-1V an admixture model was performed; it assumes that each individual has ancestry from one or more of K genetically distinct sources, and geographical information was incorporated in the analysis. For DENV-2II a recent introduction from 2017 to Peruvian territory was non-admixture assuming each individual comes purely from one of the K genetically distinct sources. To resume the replica for each K CLUMPP[26] was employed. The best K was detected and bar plots of structure analysis were carried out with Pophelper v2.3.1[27] implemented in R software.

*Bayesian phylochronological analysis* - Following a manual selection of 80 representative DENV-1 and DENV-2 genomes in terms of time (months and years) and space (provinces and countries) from the ML phylogeny, BEAST software was used to perform a Bayesian Phylochronological analysis.[28] Two separate runs were used, each comprising 50 million generations at a sample frequency of 2500 under both the strict and uncorrelated relaxed molecular clock and the Bayesian Skyline coalescent (BS) with GTR+I+G substitution models. After final outputs were examined in TRACER

and ESS values > 200 were corroborated, tree files were summarised in TreeAnnotator v1.10 with a 10% burn-in and visualised in Figtree v1.4.4.[29]

### RESULTS

*Genomic analysis* - After analysing the sequencing data of the 245 genomes reported here, we obtained an average coverage and sequencing completeness of 2241.7x and 94.7%, respectively. Since we obtained good quality information, we were able to identify a total of 100, 130, and 15 genomes respectively for DENV-1 (genotype V), DENV-2 (genotype II "Cosmopolitan"), and DENV-3 (genotype III), respectively, belonging to the 2023 outbreak. A detailed description of the mutations associated with each genotype is given in Table.

*Genetic inference and Bayesian phylochronological analysis* - The 100 DENV-1 genomes sequenced in this study belonged to genotype V and to lineages D.1 and D.2, and segregated into three main clades (I-III), each showing distinct temporal and geographical circulation patterns in Peru (Fig. 1). Clade I is spread over 15 regions of Peru and is mainly grouped with samples from north-western Brazil (cities of Rondônia and Amazonas) with common ancestry with Cuba. Clade II is composed of a few samples from northern Peru, mainly in the Tumbes region. Unlike the others, clade II has no associated mutations. Finally, clade III extends through southern and central Peru, affecting Peruvian regions such as Lima, Junin, Cusco, Madre de Dios, and Ayacucho. They are also related to samples from Brazil, Bolivia, and Colombia. Most of the Peruvian samples sequenced in the 2023 outbreak belong to clade I, followed by clades III and II, respectively (Fig. 1). The population structure analysis shows that Peru contains populations from two different clusters (cluster I and cluster II), as well as a mixture of both clusters (Fig. 1), evidencing the wide national dispersion of clade I in contrast to clade II. From this, we estimate that clades I, II, and III emerged in Brazil in 2016, which corresponds to the patterns observed in Brazilian populations. In Peru, clade I is estimated to have emerged between 2017 and 2019, clade II between 2018 and 2020, and clade III between 2019 and 2020 (Fig. 1).

The 130 genomes of DENV-2 clustered consistently within the Cosmopolitan genotype II, lineage F.1.2, forming three clades (I-III) with clear regional differentiation (Fig. 2). Clade I is made up of samples from the Amazonas, Lima, and Loreto regions, and clade II is made up of samples from Lima, Puno, Huanuco, Callao, Madre de Dios, Ayacucho, San Martin, and Ica. Clade III is evident in the northern and central regions of Peru, such as La Libertad, Piura, Tumbes, Cajamarca, Huanuco, and Ancash (Fig. 2). Although virus dispersal can be traced, the presence of biases cannot be ruled out due to the limited number of samples sequenced. Population structure analysis reveals the presence of two clusters, II and III circulating in the Peruvian samples, well established in the different regions, and samples that are a hybrid between clusters II and III are becoming prevalent over time, meaning that they could become a future well-structured clade (Fig. 2). Phylochronological analysis of the Cosmopolitan genotype supports the previously reported introduction from the Asian continent to Peru-Brazil and suggests the appearance of clade I in Peru in late 2018, spreading across most of the country and then moving to Brazil, with circulation of clade II present between 2018-2019 and clade III between 2019-2020 by 2023 (Fig. 2).

All 15 DENV-3 genomes corresponded to genotype III, lineage B.3, and clustered tightly in clade I with sequences from Cuba, Puerto Rico, Florida (USA), and Brazil, indicating a recent introduction restricted to Lima during 2023 (Fig. 3). Population structure analysis found that cases were concentrated in the Lima region, with C: I94T, NS1:S145N, and NS5:M614I mutations in the 2023 samples.

Taken together, these findings highlight both long-standing viral diversification (DENV-1, DENV-2) and recent introduction events (DENV-3III), each contributing uniquely to the complex dengue landscape in Peru.

### DISCUSSION

Since its emergence, dengue has been considered a neglected disease of global concern with an alarming progressive increase in cases.[30] Recently, in 2023, Latin American countries reported a significant increase in dengue cases. However, the lack of comprehensive genomic data has hindered efforts to understand how viral evolution influences public health. This study represents the first contribution of a large dataset of dengue genomes which allowed understanding the 2023 dengue outbreak in Peru. Among the reported cases, the DENV serotypes identified were DENV-1 (41%), DENV-2 (53%), and DENV-3 (6%), with all three circulating throughout the country. Although DENV-2 has been reported since

TABLE

Regional distribution of dengue virus (DENV) serotypes, genotypes, and associated mutations in Peru

| Serotype | Genotype | Lineage | Clade | Mutations | Peruvian regions |
|----------|----------|---------|-------|-----------|------------------|
| DENV-1 | V | D.1 | I | NS1: M178I | More than 15 regions of Peru |
| | | D.2 | III | NS2A: K218R* | Lima, Junin, Cusco, Ancash, Ayacucho, Madre de Dios, Loreto, Piura. |
| DENV-2 | II | F.1.2 | I | NS5: T176I, H247Y | More than 15 regions of Peru |
| DENV-3 | III | B.3 | I | C: I94T, NS5: M614I, NS1: S145N | Lima |

*It is not present in all samples.

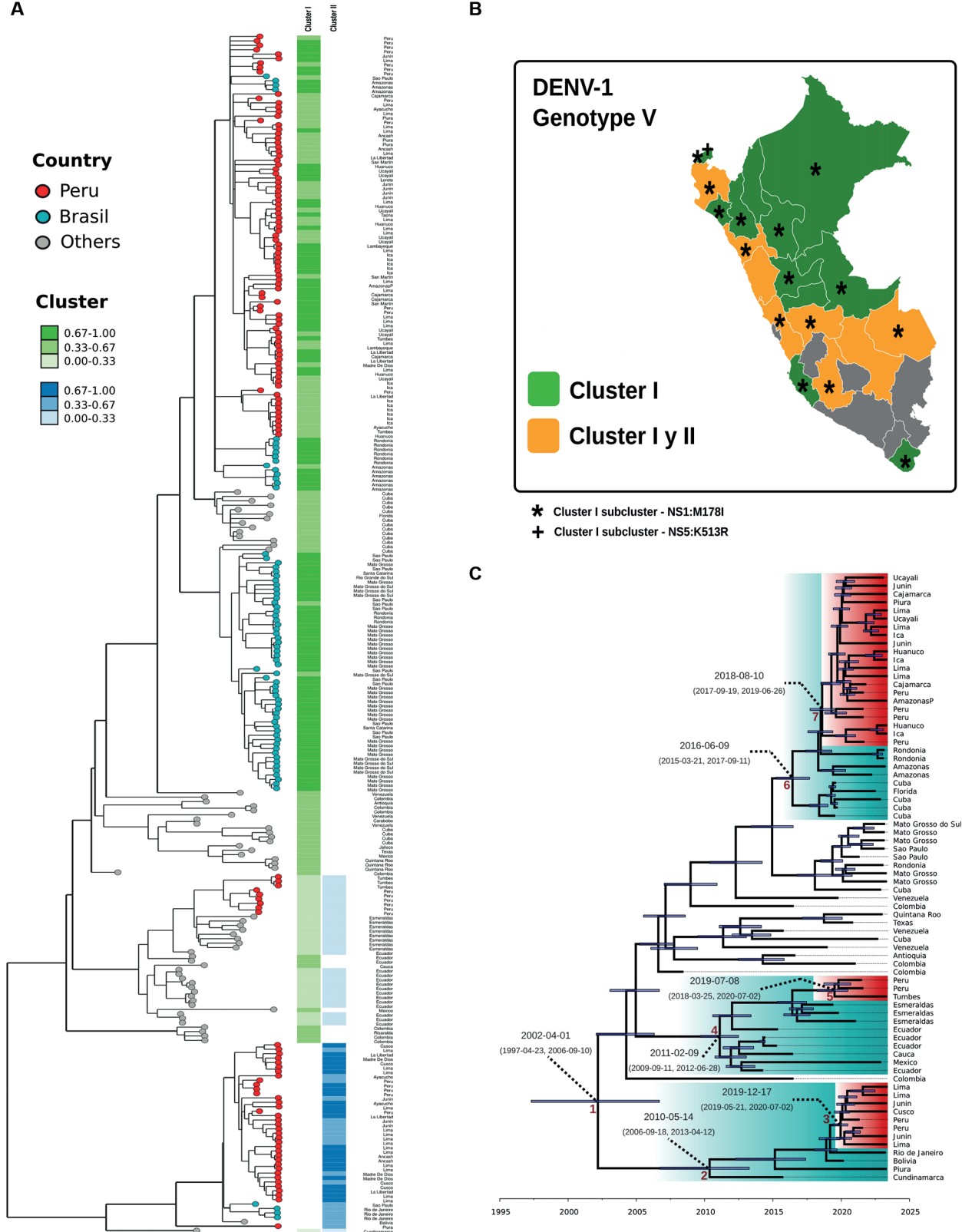

Fig. 1: genomic and phylogenetic analysis of dengue virus (DENV) serotype DENV-1 (genotype V) during the 2023 outbreak in Peru. (A) Bayesian phylogenetic clustering inferred using STRUCTURE. (B) Geographic distribution of Peruvian genomes, with symbols indicating cluster-specific mutations. (C) Bayesian phylochronological analysis based on curated, complete genomes using Bayesian evolutionary analysis by sampling trees (BEAST).

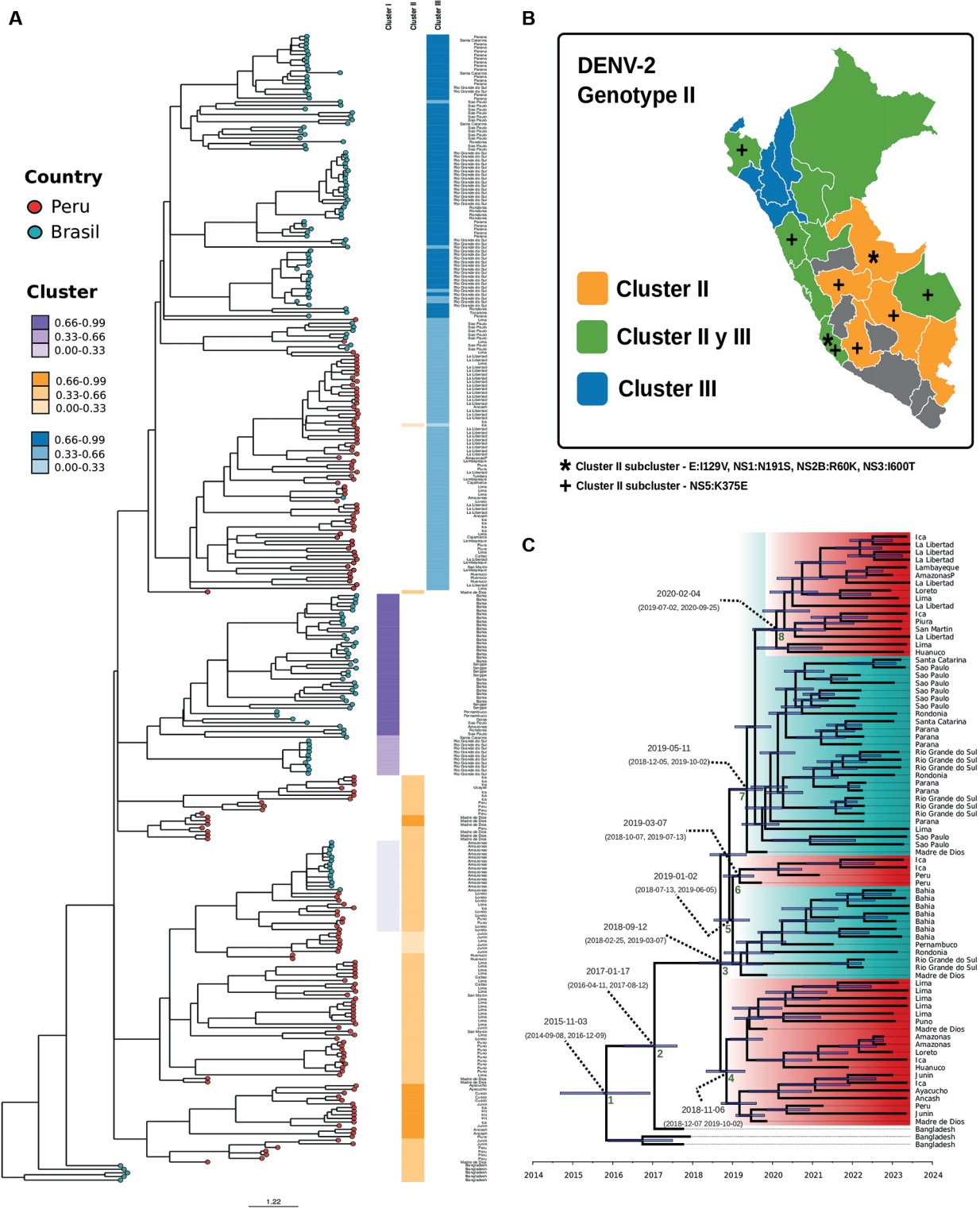

Fig. 2: genomic and phylogenetic analysis of dengue virus (DENV) serotype DENV-2 (genotype Cosmopolitan, II). (A) Bayesian phylogenetic clustering inferred using STRUCTURE. (B) Geographic distribution of Peruvian genomes, with symbols indicating cluster-specific mutations. (C) Bayesian phylochronological analysis based on curated, complete genomes using Bayesian evolutionary analysis by sampling trees (BEAST).

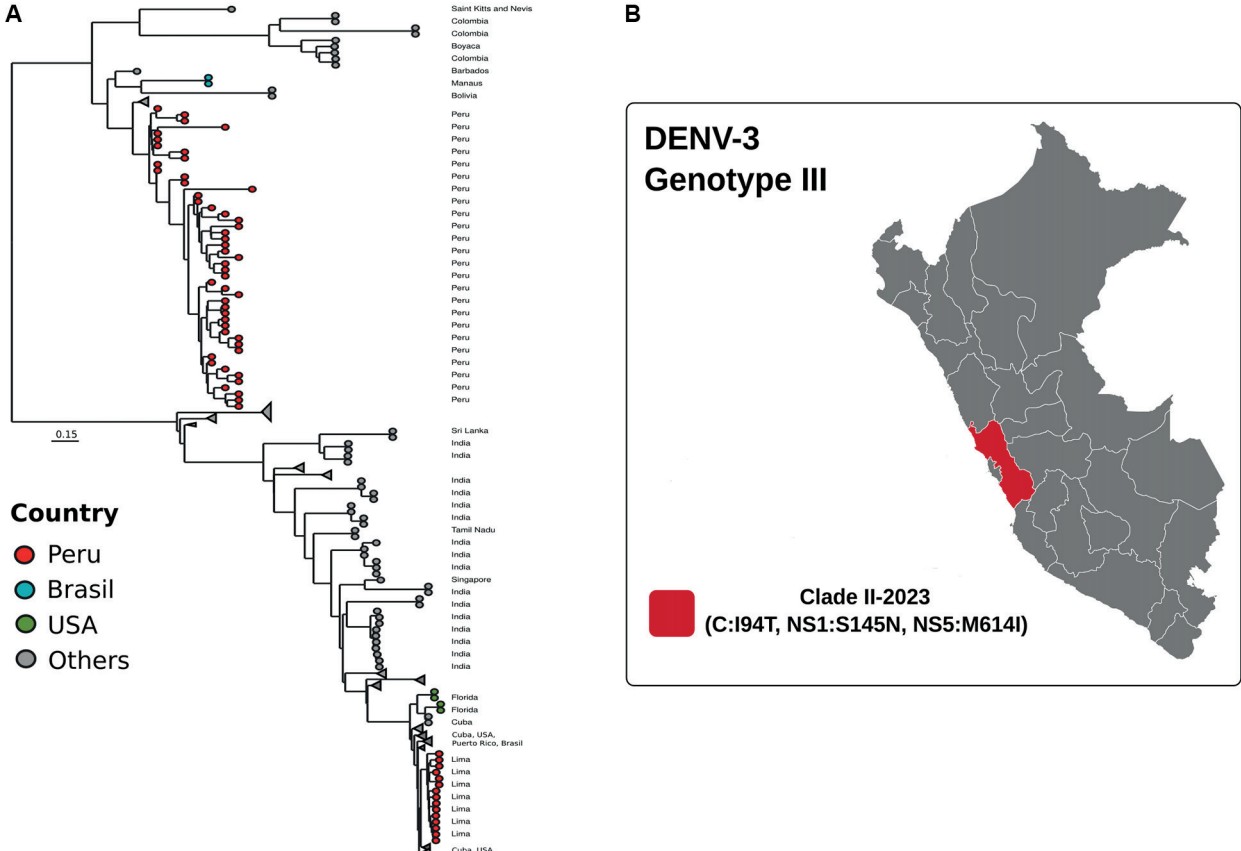

Fig. 3: genomic and phylogenetic analysis of dengue virus (DENV) serotype DENV-3 (genotype III) during the 2023 outbreak in Peru. (A) Phylogenetic tree of circulating strains in Lima. (B) Mutations identified in clade II-2023.

the 1990s with a higher prevalence, the cases with the greatest dissemination in the 2023 outbreak are those of DENV-1.[31] DENV-3 showed a low prevalence in this outbreak, being reported only in the Lima region; no cases of DENV-4 were reported. The first reports of DENV-3 in 2000 were traced in Iquitos, Lima, Tumbes, and Loreto.[32] Even though dengue serotypes have been widely reported in multiple countries, this analysis represents the first focused on the current Peruvian diversity. The massive spread of the different dengue serotypes was occurring in the various coastal, highland, and jungle regions of Peru as a presumably result of the increase in climatic factors associated with dengue due to the El Niño phenomenon that took place during that period.[33]

Given the increase in cases in the Americas, the Pan American Health Organisation (PAHO) recommended strengthening surveillance of dengue and other arboviruses, promoting their prevention, and providing access to medical care in the region.[34] For Peru, climate change and socio-demographic conditions are conducive to the development of the vector, this promotes recurrent disease which causes outbreaks annually, including the last one reported in 2023.

DENV-1, according to the results reported, belongs to genotype V, which is prevalent in a single dispersal route in Pacific countries; these findings are supported by previous studies using partial E and complete gene information.[35] The most divergent clade was detected in genomes published in the GISAID database from the outbreak in 2023 in Brazil, suggesting ancestral relationships with Venezuela and Argentina, a likely route of dispersal from Atlantic countries by export.[36] These results support the evidence that genetic drift caused the geographic spread of DENV-1 V genotype that occurred in South America. Inside the lineages reported, D.1 and D.2, the last one is less distributed and has only been reported from South American countries such as Colombia, Bolivia, Brazil, and Peru.[6] The evolution and divergence of each clade occurred in continuous outbreaks; each clade has a particular mutation profile (Table), and it is possible to detect multiple exports of Peruvian clades to Brazil.[9] Clade I was exported to Amazonas and Rondônia, and Clade III was introduced to São Paulo and Rio de Janeiro. However, this introduction reported low epidemiological significance related to the 2023 outbreak in Brazil. Bayesian population analysis finds K = 2, allowing reconstruction of the genetic history into two clusters I and II (Fig. 1), but ancestry reconstruction shows individuals from 100% cluster I, others from 100% cluster II, and others as hybrids. Careful interpretation of the results is therefore necessary due to the lack of complete genomic samples in the past for dengue. The possible multiple introductions found in the phylogenetics from Peru to Brazil can be evidenced because they

share individuals with cluster II mainly. Cluster I and hybrid individuals were ancestral sequences from Cuba, Ecuador, Colombia, Venezuela, and Mexico.[25,28]

In DENV-2, analysis identified all samples from outbreak 2023 as Cosmopolitan (genotype II), suggesting that the main introduction of Dengue Cosmopolitan to Peru was from Bangladesh in September 2018; these findings confirm previous reports using partial envelope gene sequences.[10] In Peru, there was an outbreak in 1995 in the cities of Iquitos, Pucallpa, and three other cities in northern Peru.[37] An active circulation of the Asian/American genotype was confirmed in the cities of Madre de Dios and Loreto from 2002 until 2018.[10,38] Currently, a large outbreak shows the prevalence of Cosmopolitan over other genotypes, and its spread throughout Peruvian territories suggests genotype replacement. We identified the circulation of the F.1.2 lineage of DENV-2II in our samples from outbreak 2023, which spread mainly in Southeast and East Asia since 2009. These results reveal that the Cosmopolitan F.1.2 lineage was circulating in Asia almost a decade before its introduction in Peru in 2018.[6] Phylogenetic analysis shows the export of the Peruvian cosmopolitan to Brazil on multiple occasions, and dispersal patterns like those of DENV-1 can be described. A zone of interaction between the jungle borders and other direct spread to Porto Alegre, São Paulo, and Paraná is supported by research conducted in the 2023 outbreak in Brazil.[39] These interaction zones were reported for Asia/Americas in the 2009-2010 Peruvian outbreak but probably from Brazil to Peru,[40] the sparse genomic information suggests dispersal. In addition, population structure analyses show that clusters I, and II are ancestrally related to Bangladesh, however, cluster III has a new genomic composition. The Cosmopolitan genotype of DENV-2 reported in Peru has a more complex mixed ancestry composition. Our research suggests more than one Cosmopolitan introduction existed in the past in our Peruvian territory. On the other hand, we have not observed a strong structure in the Peruvian population, probably due to sampling bias, but it is very useful to understand the multiple introductions to Brazil, Amazonas, and São Paulo and two other populations of pure cluster I or cluster III individuals. It is interesting to contrast the well-structured population in Brazil and the diffuse clustering pattern observed in Peru, hypothesising that a new population structure may soon be defined. Likewise, the clusters observed in Peru seem to be distributed in different parts of the country, showing a higher prevalence of cluster II in southeastern Peru and cluster III in northern Peru, evidently showing a mixture of them in the middle zones of interconnection. Finally, the emergence of an epidemiologically important clade of Cosmopolitan was detected in the large outbreak of 2023, and genomic surveillance is urgently needed to understand the evolution of DENV-2 cosmopolitan. In this regard, DENV-3 is the least prevalent type of dengue in Peru and reports to the Peruvian Ministry of Health on the spread of DENV-3 in South America conclude that most of the viruses circulating in South America have evolved and formed phylogenetic groups distinct from those in Central America.[41] In March 2023, the

mean weekly number of dengue cases in Peru increased sharply, from 2,182 during epidemiologic weeks 1-10 (corresponding to January 1-March 11) to 8,787 during weeks 11-20 (March 12-May 20). As of the end of week 30 (July 29), the 222,620 cases in 2023 were approximately 10 times the average number during the same period during the previous 5 years (21,841 cases) and 3.5 times the number during the same period in 2017 (64,431 cases), the year of the largest previous national dengue outbreak.[42] Our analysis confirms that genotype III genomes from a large outbreak in 2023 have phylogenetic relationships with recent circulating viruses present in Cuba and Florida-USA (Fig. 3). During 2024, DENV-3 circulation was identified in the regions of Lima, Loreto, San Martin, Piura, Cajamarca, Amazonas, Ancash, Ica, Callao, Ayacucho, Huanuco, and Ucayali, where previously only DENV-1 and DENV-2 had been reported.[43]

In 2023, the detection of DENV-3III was limited exclusively to the city of Lima, representing a localised transmission event with clearly defined temporal and geographical relevance. The identified B.3 lineage showed phylogenetic affinity with sequences circulating contemporaneously in the Caribbean and the southeastern United States, suggesting a recent introduction within an active regional epidemiological corridor. Although circulation in 2023 was focal, the subsequent nationwide dissemination documented during 2024 demonstrates that this lineage later became established across multiple regions of Peru. These patterns underscore the importance of continuous genomic surveillance to identify early introductions that, as in this case, may precede broad-scale spread in subsequent years.

The lineage detected in Peru was B.3, a lineage widely distributed in the Americas, Africa, and Asia. However, it is most prevalent in North America, Central America, and the Caribbean. Although the literature reports that genotype III has been circulating in Peru since 2002[44] and there are Peruvian public genomes between 2002 and 2009 in the NCBI database, they do not share a related evolutionary history. Therefore, the genomes reported for the 2023 outbreak were probably a case of importation, because all isolates were from the Peruvian capital, Lima. We consider it crucial to focus our attention on genomic surveillance to understand the dynamics of DENV-1, DENV-2, and DENV-3 genotypes in Peru, including DENV-4 even though it has not been reported since the last decade in the different endemic areas impacting active surveillance.[45] Peru must reinforce its efforts to make surveillance considering the high prevalence of the disease and its wide recurrence annually, sharing the diversity of genomic information meaning a significant contribution to the scientific community. Presumably, the clinical samples received at INS for surveillance significantly underestimate the true circulation of DENV-3 genotype III in these studies, and suggest necessary an active surveillance. For this reason, the genomic data from the large Peruvian outbreak in 2023 show a more complex scenario. We propose to improve the full recognition of the DENV to understand the dynamics and diversity of genotypes and the new clades (sub-lineages) detected within each genotype.

The genomic surveillance data obtained on circulating dengue genotypes would provide us with useful information on genotype dynamics in our country, which should be taken into account for the implementation of control measures such as prioritising vaccination in at-risk groups or regions with dynamic genotype change. Furthermore, the integration of dengue genomic surveillance data with entomological and climatological data would be useful for addressing effective and prioritised vector control at the regional and local levels.

This study provides the most comprehensive genomic overview of dengue circulation during the 2023 outbreak in Peru. We demonstrate that DENV-1 and DENV-2 exhibited multiple introductions and extensive nationwide dispersal, whereas DENV-3III represented a localised introduction in Lima with later expansion in 2024. These results underscore the need for sustained, real-time genomic surveillance to detect emerging lineages and guide targeted public health interventions.

## ACKNOWLEDGEMENTS

To all the Regional Reference Laboratories in Peru that collaborate with the Metaxenic and Viral Zoonoses Laboratory of the National Centre for Public Health at the National Institute of Health in Peru by sending samples suspected of dengue.

## AUTHORS' CONTRIBUTION

OM and VJ-V - writing, analysis, review and design of all experiments; WL - writing, review and design; PM, IS-M, NS, VH, LB-F, VI-R, AN-L, KV-C, HB, MG-P, SM, DF and AM - sampling, review and experimental processing; CP-R and MPG - coordination and review. All authors have review and approve the final manuscript. The authors declare no conflict of interest.

## DATA AVAILABILITY

Sequence data and metadata that support the findings of this article have been deposited in the GISAID database with accession ID for DENV-1 (EPI_ISL_20098504 - EPI_ISL_20098609), DENV-2 (EPI_ISL_20098387 - EPI_ISL_20098503), and DENV-3 (EPI_ISL_20098610 - EPI_ISL_20098624).

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

# OPEN PEER REVIEW

Memórias do IOC thanks the anonymous reviewers for their contribution to the peer review of this work.

## FIRST REVIEW ROUND

### REVIEWERS' COMMENTS

**REVIEWER #1**

This manuscript presents genomic surveillance findings from the 2023 dengue outbreak in Peru, analyzing 238 high-quality viral genomes and characterizing DENV-1 genotype V (D.1/D.2), DENV-2 Cosmopolitan (II_F.1.2), and DENV-3III (B.3). The dataset is the largest Peru-focused collection reported to date and represents an important contribution by adopting the 2024 dengue lineage nomenclature and applying population-structure analysis, which is rarely used in dengue studies. The study is well executed and provides valuable insights into lineage diversity, introductions, and dispersal patterns, with clear implications for public health surveillance.

However, several revisions are needed before publication. The novelty claims should be moderated, as Peru–Brazil viral flow for DENV-2 Cosmopolitan and lineage structure of DENV-1V have been reported previously; the contribution here lies in dataset scale, Peru-centric analysis, and added methods. The discussion of DENV-3III should stress the local and temporal significance (Lima 2023) while acknowledging contemporaneous circulation in the Caribbean and subsequent Peru-wide spread in 2024. Some references are weak or misaligned, notably Ref 21 (fish virus genetics), Ref 34 (East Africa DENV-2), and Ref 17 (Zika paper cited for IVAR parameters); these should be replaced or supplemented with dengue-specific sources. Key numerical claims (Ref 9 on "367 genomes," Ref 13 on outbreak case counts) should be verified against PAHO or MOH reports. The discussion would benefit from clearer explanation of sample representativeness, justification of STRUCTURE assumptions, and more accessible figure/table legends that highlight the practical significance of clades and mutations. Finally, public health implications should be emphasized more explicitly, for example by suggesting how genomic surveillance can inform outbreak response, vaccination planning, or integration with climate and entomological data.

In summary, this study provides a valuable dataset and analysis of the 2023 dengue outbreak in Peru, but major revisions are required to moderate claims, strengthen citations, and enhance clarity and interpretive depth.

### AUTHORS' RESPONSE TO THE REVIEWERS

I have reviewed carefully the comments of the two experts described in the previous email in reference to the manuscript "Molecular epidemiology of large dengue serotypes outbreak from Peru, 2023" with Manuscript ID MIOC-2025-0204 and below I explained the modifications and corrections made in each detailed point:

Reviewer 1

A) The novelty claims should be moderated, as Peru–Brazil viral flow for DENV-2 Cosmopolitan and lineage structure of DENV-1V have been reported previously; the contribution here lies in dataset scale, Peru-centric analysis, and added methods.

Reply to the reviewer:

Based on the suggestion, we have made modifications to the discussion, adding the following: "This study represents the first contribution of a large dataset of dengue genomes which allowed understanding the 2023 dengue outbreak in Peru" (lines 196-198), "Even though dengue serotypes have been widely reported in multiple countries, this analysis represents the first focused on the current Peruvian diversity" (lines 204-205), "this promotes recurrent disease which causes outbreaks annually, including the last one reported in 2023" (lines 212-213), "Inside the lineages reported, D.1 and D.2, the last one is" (line 220-221);"Perú must reinforce its efforts to make surveillance considering the high prevalence of the disease and its wide recurrence annually, sharing the diversity of genomic information meaning a significant contribution to the scientific community." (line 297-299).

B) The discussion of DENV-3III should stress the local and temporal significance (Lima 2023) while acknowledging contemporaneous circulation in the Caribbean and subsequent Peru-wide spread in 2024.

Reply to the reviewer:

Based on the suggestion, We added additional information on dengue circulation in Peru by epidemiological weeks (lines 268-273) and the appearance of serotypes 1, 2, and 3 in different departments of Peru (lines 275-278). In addition, we have updated the references (39 and 40 respectively).

C) Some references are weak or misaligned, notably Ref 21 (fish virus genetics), Ref 34 (East Africa DENV-2), and Ref 17 (Zika paper cited for IVAR parameters); these should be replaced or supplemented with dengue-specific sources.

Reply to the reviewer:

Based on the suggestion, we have updated the references, about the Ref 21 was replaced by the new reference on line 384: Suchard MA, Lemey P, Baele G, Ayres DL, Drummond AJ, Rambaut A. Bayesian phylogenetic and phylodynamic data integration using BEAST 1.10. Virus Evol. 2018;4(1):vey016. The Ref 34 was removed on line 419. And the Ref 17 was replaced by the new reference on line 372: Grubaugh ND, Gangavarapu K, Quick J, Matteson NL, Oliveira G, et al. An amplicon-based sequencing framework for accurately measuring intrahost virus diversity using PrimalSeq and iVar. Genome Biol. 2019;20(1):8. doi:10.1186/s13059-018-1618-7.

D) Key numerical claims (Ref 9 on "367 genomes," Ref 13 on outbreak case counts) should be verified against PAHO or MOH reports.

Reply to the reviewer:

After reviewing the GenBank database through NCBI Virus, we found 1291 dengue genomes with >70% coverage deposited up to 2026. However, we cannot verify the accuracy of the "367 genomes" indicated in the reference, as we lack information on when they used the cutoff date for their analysis; we only know that the article was published in 2023. We found no reports from the WHO or PAHO indicating the number of shared genomes. Therefore, we consider it appropriate to retain reference 9, as it is possible that they did find that number of genomes due to the cutoff date of their analysis.

We thank the reviewer for the observation in ref 13. The information on the number of dengue cases in the outbreak in Peru has been updated in the corresponding lines 71-73, and the reference has been replaced with the official PAHO report. The link and full citation are provided; Organización Panamericana de la Salud / Organización Mundial de la Salud. Actualización Epidemiológica: Dengue, chikunguña y Zika. Washington, D.C.: OPS/OMS; 2023 Jun 10 .

E) The discussion would benefit from clearer explanation of sample representativeness, justification of STRUCTURE assumptions, and more accessible figure/table legends that highlight the practical significance of clades and mutations.

Reply to the reviewer:

Dear reviewer. The titles of the figures and table have been improved.

Related to sample representative: We acknowledge that our sample collection was opportunistic rather than part of a systematic design, as obtaining high-quality RNA samples from remote areas of Peru presented significant logistical challenges. Consequently, our discussion focuses on the robust characterization of the successfully sequenced lineages, rather than on extrapolating their precise geographic distribution or frequency across Peru.

Justification of STRUCTURE assumptions: In Materials and methods, in the Inference of genetic structure we explain the assumptions for Structure analysis. (124-128) "... For DENV-1V an admixture model was performed; it assumes that each individual has ancestry from one or more of K genetically distinct sources, and geographical information was incorporated in the analysis. For DENV-2II a recent introduction from 2017 to Peruvian territory was non-admixture assuming each individual comes purely from one of the K genetically distinct sources".

Explain: For DENV-1 Genotype V, which has a longer and more complex circulation history in South America, we used the admixture model with correlated allele frequencies. This model allows for mixed ancestry and is appropriate for populations with high mutation rates and potential recombination between sub-lineages or multiple introduction events. And, the DENV-2 Cosmopolitan Genotype, whose introduction to Peru is documented as a recent, discrete event (circa 2017-2018), we used the no-admixture model.

F) Finally, public health implications should be emphasized more explicitly, for example by suggesting how genomic surveillance can inform outbreak response, vaccination planning, or integration with climate and entomological data.

Reply to the reviewer:

We appreciate your suggestion; we have included a paragraph in the discussion to more explicitly emphasize how genomic surveillance can inform outbreak response, vaccination planning, and integration with climatic and entomological data: "The genomic surveillance data obtained on circulating dengue genotypes would provide us with useful information on genotype dynamics in our country, which should be taken into account for the implementation of control measures such as prioritizing vaccination in at-risk groups or regions with dynamic genotype change. Furthermore, the integration of dengue genomic surveillance data with entomological and climatological data would be useful for addressing effective and prioritized vector control at the regional and local levels".

We attach the new version of the manuscript, highlighting the changes requested in red and some minor changes made by authors in blue. In addition, please let me know if any point is not completely clear. I would like to mention that we are completely grateful for your time and consideration. We hope to hear from you soon.

Best regards,

## SECOND REVIEW ROUND

REVIEWERS' COMMENTS

### REVIEWER #1

No other comments.

