## [Reviewer Report · FIRST REVIEW ROUND - REVIEWERS COMMENTS]

## REVIEWER #1

This manuscript presents genomic surveillance findings from the 2023 dengue outbreak in Peru, analyzing 238 high-quality viral genomes and characterizing DENV-1 genotype V (D.1/D.2), DENV-2 Cosmopolitan (II_F.1.2), and DENV-3III (B.3).

The dataset is the largest Peru-focused collection reported to date and represents an important contribution by adopting the 2024 dengue lineage nomenclature and applying population-structure analysis, which is rarely used in dengue studies.

The study is well executed and provides valuable insights into lineage diversity, introductions, and dispersal patterns, with clear implications for public health surveillance.

However, several revisions are needed before publication. The novelty claims should be moderated, as Peru–Brazil viral flow for DENV-2 Cosmopolitan and lineage structure of DENV-1V have been reported previously; the contribution here lies in dataset scale, Peru-centric analysis, and added methods.

The discussion of DENV-3III should stress the local and temporal significance (Lima 2023) while acknowledging contemporaneous circulation in the Caribbean and subsequent Peru-wide spread in 2024. Some references are weak or misaligned, notably Ref 21 (fish virus genetics), Ref 34 (East Africa DENV-2), and Ref 17 (Zika paper cited for IVAR parameters); these should be replaced or supplemented with dengue-specific sources. Key numerical claims (Ref 9 on “367 genomes,” Ref 13 on outbreak case counts) should be verified against PAHO or MOH reports.

The discussion would benefit from clearer explanation of sample representativeness, justification of STRUCTURE assumptions, and more accessible figure/table legends that highlight the practical significance of clades and mutations.

Finally, public health implications should be emphasized more explicitly, for example by suggesting how genomic surveillance can inform outbreak response, vaccination planning, or integration with climate and entomological data.

In summary, this study provides a valuable dataset and analysis of the 2023 dengue outbreak in Peru, but major revisions are required to moderate claims, strengthen citations, and enhance clarity and interpretive depth.

## AUTHORS’ RESPONSE TO THE REVIEWERS

I have reviewed carefully the comments of the two experts described in the previous email in reference to the manuscript “Molecular epidemiology of large dengue serotypes outbreak from Peru, 2023” with Manuscript ID MIOC-2025-0204 and below I explained the modifications and corrections made in each detailed point:

**Reviewer 1**

**A) The novelty claims should be moderated, as Peru–Brazil viral flow for DENV-2 Cosmopolitan and lineage structure of DENV-1V have been reported previously; the contribution here lies in dataset scale, Peru-centric analysis, and added methods.**

*Reply to the reviewer*:

Based on the suggestion, we have made modifications to the discussion, adding the following: “This study represents the first contribution of a large dataset of dengue genomes which allowed understanding the 2023 dengue outbreak in Peru” (lines 196-198), “Even though dengue serotypes have been widely reported in multiple countries, this analysis represents the first focused on the current Peruvian diversity” (lines 204-205), “this promotes recurrent disease which causes outbreaks annually, including the last one reported in 2023” (lines 212-213), “Inside the lineages reported, D.1 and D.2, the last one is” (line 220-221),“Perú must reinforce its efforts to make surveillance considering the high prevalence of the disease and its wide recurrence annually, sharing the diversity of genomic information meaning a significant contribution to the scientific community.” (line 297-299).

**B) The discussion of DENV-3III should stress the local and temporal significance (Lima 2023) while acknowledging contemporaneous circulation in the Caribbean and subsequent Peru-wide spread in 2024.**

*Reply to the reviewer*:

Based on the suggestion, We added additional information on dengue circulation in Peru by epidemiological weeks (lines 268-273) and the appearance of serotypes 1, 2, and 3 in different departments of Peru (lines 275-278).

In addition, we have updated the references (39 and 40 respectively).

**C) Some references are weak or misaligned, notably Ref 21 (fish virus genetics), Ref 34 (East Africa DENV-2), and Ref 17 (Zika paper cited for IVAR parameters); these should be replaced or supplemented with dengue-specific sources.**

*Reply to the reviewer*:

Based on the suggestion, we have updated the references, about the Ref 21 was replaced by the new reference on line 384: Suchard MA, Lemey P, Baele G, Ayres DL, Drummond AJ, Rambaut A. Bayesian phylogenetic and phylodynamic data integration using BEAST 1.10. Virus Evol. 2018;4(1):vey016. The Ref 34 was removed on line 419. And the Ref 17 was replaced by the new reference on line 372: Grubaugh ND, Gangavarapu K, Quick J, Matteson NL, Oliveira G, et al. An amplicon-based sequencing framework for accurately measuring intrahost virus diversity using PrimalSeq and iVar. Genome Biol. 2019;20(1):8. doi:10.1186/s13059-018-1618-7.

**D) Key numerical claims (Ref 9 on “367 genomes,” Ref 13 on outbreak case counts) should be verified against PAHO or MOH reports.**

*Reply to the reviewer*:

After reviewing the GenBank database through NCBI Virus, we found 1291 dengue genomes with >70% coverage deposited up to 2026. However, we cannot verify the accuracy of the “367 genomes” indicated in the reference, as we lack information on when they used the cutoff date for their analysis; we only know that the article was published in 2023. We found no reports from the WHO or PAHO indicating the number of shared genomes.

Therefore, we consider it appropriate to retain reference 9, as it is possible that they did find that number of genomes due to the cutoff date of their analysis.

We thank the reviewer for the observation in ref 13. The information on the number of dengue cases in the outbreak in Peru has been updated in the corresponding lines 71-73, and the reference has been replaced with the official PAHO report.

The link and full citation are provided; Organización Panamericana de la Salud / Organización Mundial de la Salud. Actualización Epidemiológica: Dengue, chikunguña y Zika. Washington, D.C.: OPS/OMS; 2023 Jun 10 .

**E) The discussion would benefit from clearer explanation of sample representativeness, justification of STRUCTURE assumptions, and more accessible figure/table legends that highlight the practical significance of clades and mutations.**

*Reply to the reviewer*:

Dear reviewer. The titles of the figures and table have been improved.

Related to sample representative: We acknowledge that our sample collection was opportunistic rather than part of a systematic design, as obtaining high-quality RNA samples from remote areas of Peru presented significant logistical challenges. Consequently, our discussion focuses on the robust characterization of the successfully sequenced lineages, rather than on extrapolating their precise geographic distribution or frequency across Peru.

Justification of STRUCTURE assumptions: In Materials and methods, in the Inference of genetic structure we explain the assumptions for Structure analysis.

(124-128) “... For DENV-1V an admixture model was performed; it assumes that each individual has ancestry from one or more of K genetically distinct sources, and geographical information was incorporated in the analysis. For DENV-2II a recent introduction from 2017 to Peruvian territory was non-admixture assuming each individual comes purely from one of the K genetically distinct sources”.

Explain: For DENV-1 Genotype V, which has a longer and more complex circulation history in South America, we used the admixture model with correlated allele frequencies. This model allows for mixed ancestry and is appropriate for populations with high mutation rates and potential recombination between sub-lineages or multiple introduction events.

And, the DENV-2 Cosmopolitan Genotype, whose introduction to Peru is documented as a recent, discrete event (circa 2017-2018), we used the no-admixture model.

**F) Finally, public health implications should be emphasized more explicitly, for example by suggesting how genomic surveillance can inform outbreak response, vaccination planning, or integration with climate and entomological data.**

*Reply to the reviewer*:

We appreciate your suggestion; we have included a paragraph in the discussion to more explicitly emphasize how genomic surveillance can inform outbreak response, vaccination planning, and integration with climatic and entomological data: “The genomic surveillance data obtained on circulating dengue genotypes would provide us with useful information on genotype dynamics in our country, which should be taken into account for the implementation of control measures such as prioritizing vaccination in at-risk groups or regions with dynamic genotype change. Furthermore, the integration of dengue genomic surveillance data with entomological and climatological data would be useful for addressing effective and prioritized vector control at the regional and local levels”.

We attach the new version of the manuscript, highlighting the changes requested in red and some minor changes made by authors in blue.

In addition, please let me know if any point is not completely clear.

I would like to mention that we are completely grateful for your time and consideration.

We hope to hear from you soon.

Best regards,